# Semi-Supervised Domain Generalization with Known and Unknown Classes

**Lei Zhang, Ji-Fu Li, Wei Wang**[*]

National Key Laboratory for Novel Software Technology, Nanjing University, China
School of Artificial Intelligence, Nanjing University, China
`{zhangl,lijf,wangw}@lamda.nju.edu.cn`

## Abstract

Semi-Supervised Domain Generalization (SSDG) aims to learn a model that is generalizable to an unseen target domain with only a few labels, and most existing SSDG methods assume that unlabeled training and testing samples are all known classes. However, a more realistic scenario is that known classes may be mixed with some unknown classes in unlabeled training and testing data. To deal with such a scenario, we propose the Class-Wise Adaptive Exploration and Exploitation (CWAEE) method. In particular, we explore unlabeled training data by using one-vs-rest classifiers and class-wise adaptive thresholds to detect known and unknown classes, and exploit them by adopting consistency regularization on augmented samples based on Fourier Transformation to improve the unseen domain generalization. The experiments conducted on real-world datasets verify the effectiveness and superiority of our method.

## 1 Introduction

The machine learning community has witnessed the great progress of deep learning models and their applications, e.g., computer vision [1, 2], natural language processing [3, 4]. Such a huge success is mostly based on the i.i.d. assumption, i.e., the training data and testing data are identically and independently distributed. However, when these popular models are evaluated on new testing data, whose distribution is slightly different from the training data, a significant drop in performance will be observed [5, 6].

In order to improve the generalization of the model under distribution shifts, *Domain Adaptation* (DA) [7, 8, 9] and *Domain Generalization* (DG) [10, 11, 12] have been widely studied. The goal of DA is to transfer the knowledge learned

Figure 1: Semi-supervised domain generalization with known and unknown classes. Labeled samples on source domains (Art, Cartoon and Photo) are *known classes* (green box). *Known classes* (green box) and *seen unknown classes* (blue box) are mixed in unlabeled samples on source domains, while *known classes* (green box), *seen unknown classes* (blue box) and *unseen unknown classes* (red box) are mixed in testing samples on target domain (Sketch).

---

[*]Corresponding author.

37th Conference on Neural Information Processing Systems (NeurIPS 2023).

from label-rich source domains to the unlabeled or partially labeled target domain. However, the accessibility of the target domain may not always be satisfied in some applications, e.g., autonomous driving and medical diagnosis, since we are not able to anticipate the domain that deployed systems will encounter. To this end, DG is introduced to develop domain-generalizable models on *unseen* target domain by using multiple different and labeled source domains.

To reduce the expensive cost of the labeling process [13], Semi-Supervised Domain Generalization (SSDG) [14, 15, 16, 17] is proposed recently. In SSDG, each source domain consists of a few labeled samples and a large number of unlabeled samples. The goal is to learn a domain-generalizable model from the partially labeled samples. However, in a more realistic scenario, *known classes* are probably mixed with some *unknown classes* in unlabeled training and testing data (Figure 1). In such a scenario, when deployed on an *unseen* target domain, the learned model is not only required to classify *known classes*, but also to recognize *unknown classes*. In this paper, we propose the Class-Wise Adaptive Exploration and Exploitation (CWAEE) method for semi-supervised domain generalization when *known* and *unknown classes* are mixed in unlabeled data. The intuition is to first explore unlabeled training data by detecting *known* and *unknown classes*, and then exploit them in different ways to improve the generalization of the model. In particular, we use one-vs-rest classifiers and class-wise adaptive thresholds to detect *known* and *unknown classes* in unlabeled training data, and adopt consistency regularization on augmented samples based on Fourier Transformation to improve the *unseen* domain generalization. The experiments on real-world datasets verify that our CWAEE method can achieve better performance than compared methods.

## 2   Related Work

Domain Generalization (DG) aims to help the model generalize to an unseen target domain by utilizing one or several related domains. Most existing DG methods can be classified into three different categories: domain-invariant representation learning, meta-learning and data augmentation. Motivated by the learning theory of domain adaptation [18], domain-invariant representation learning methods attempt to learn a representation space where the discrepancy between different source domains is as small as possible. For example, Muandet *et al.* [11] proposed a kernel-based optimization algorithm for Support Vector Machine that learns an invariant transformation by minimizing the distribution discrepancy across domains. After the domain-adversarial neural network was proposed for domain adaptation [8], domain-adversarial training is widely used for DG [19, 20]. Following Model-Agnostic Meta-Learning (MAML) [21], Li *et al.* [22] applied meta-learning strategy to DG through dividing the training data from multiple source domains into meta-train and meta-test sets to simulate domain shifts. The meta-optimization objective was designed to minimize the loss not only on meta-train domains but also on the meta-test domain. Shu *et al.* [23] conducted meta-learning over augmented domains to learn open-domain generalizable representations for Open Domain Generalization (OpenDG) problem. Compared to domain-invariant representation learning and meta-learning, data augmentation methods have shown better performance on various DG benchmarks recently. Zhou *et al.* [24] mixed feature statistics extracted by bottom layers of Convolutional Neural Networks (CNNs) between instances of different domains to synthesize instances in novel domains for training. Xu *et al.* [25] applied MixUp [26] in amplitude spectrums of two images to augment training data while preserving high-level semantics of the original images in phase spectrums. To reduce the labeling cost of DG, Semi-Supervised Domain Generalization (SSDG) [14, 15, 16, 17] receives much attention recently. Sharifi-Noghabi *et al.* [14] proposed the first SSDG method which combines the pseudo-label method and meta-learning strategy. Zhou *et al.* [16] adopted stochastic classifier and multi-view consistency to improve the quality of pseudo-labels and generalization of model respectively. However, they assumed that both unlabeled training and testing data have the same label space with labeled training data while we consider a more realistic scenario that the label space of labeled data is a subset of unlabeled training and testing data, i.e., both unlabeled training and testing data are a mixture of *known* and *unknown classes*.

Out-of-Distribution (OOD) detection aims to detect test samples whose labels are not contained in the label space of training data [27]. Hendrycks and Gimpel [28] proposed the first effective baseline that uses the maximum softmax probability to identify OOD samples. Post-hoc methods attract much attention because they can be implemented without modifying the training procedure and objective. For example, Liang *et al.* [29] used temperature scaling and input perturbations to make In-Distribution (ID) and OOD samples more separable. Liu *et al.* [30] proposed to replace softmax

confidence scores with energy scores to detect OOD samples, and considered samples with higher energy to be OOD samples. Hendrycks *et al.* [31] maximized the entropy of the model's predictions on enormous unlabeled OOD samples to improve OOD detection. Yang *et al.* [32] proposed the Unsupervised Dual Grouping (UDG) method that adopts two classification heads to dynamically separate ID and OOD samples in the unlabeled data and optimizes the model with different learning objectives. Since the commonly-used OOD detection benchmarks [28, 29, 31] are based on dataset splitting setup, i.e., one dataset as ID and all the others as OOD, the existing OOD detection methods overfit to low-level dataset statistics instead of learning semantics, which leads to that even the same class from another dataset will be detected as OOD samples [32].

Semi-Supervised Learning (SSL) is an active research area in the last decade, whose goal is to leverage both labeled and unlabeled data to improve the performance of models and reduce the label cost. The most popular methods include consistency regularization [33, 34] and entropy minimization [35, 36]. The basic idea behind consistency regularization is that the model's outputs of similar samples should be similar. Sajjadi *et al.* [33] proposed the $\Pi$ model that minimizes consistency loss over two random augmentations of one sample to regularize the model. Tarvainen *et al.* [34] proposed the Mean Teacher method that averages Student model weights using Exponential Moving Average (EMA) over training steps as Teacher model to provide more stable targets for the consistency loss optimization of the Student model. Entropy minimization methods encourage models to make low entropy predictions on unlabeled data, so that the learned decision boundary passes through a low-density region rather than a high-density area. Lee [36] proposed the Pseudo-label method that sets the maximum confidence prediction of unlabeled samples as pseudo-labels, and then trains the model in a supervised way with labeled and pseudo-labeled data. Sohn *et al.* [37] proposed the FixMatch method that combines consistency regularization and entropy minimization. It generates highly confident pseudo-labels on weakly-augmented unlabeled samples, and trains the model to predict the pseudo-labels for the strongly-augmented version of the same samples. OpenMatch [38] extended FixMatch with OVANet [39] and open-set soft-consistency regularization to detect outliers in unlabeled data for Open-set Semi-Supervised Learning (OSSL).

# 3 Method

We have multiple source domains $\mathcal{D}_1, \mathcal{D}_2, \ldots, \mathcal{D}_s$ available for training, and each $\mathcal{D}_{k \in \{1,2,\ldots,s\}}$ has one labeled set $\mathcal{D}_k^l = \{(\mathbf{x}_i^{(k)}, y_i^{(k)})\}_{i=1}^{n_k^l}$ and one unlabeled set $\mathcal{D}_k^u = \{(\mathbf{u}_i^{(k)})\}_{i=1}^{n_k^u}$, i.e., $\mathcal{D}_k = \mathcal{D}_k^l \cup \mathcal{D}_k^u$. Generally, the size of $\mathcal{D}_k^u$ may be much larger than that of $\mathcal{D}_k^l$, i.e., $n_k^u \gg n_k^l$. The total size of labeled dataset is denoted as $n^l$ and the total size of unlabeled dataset is denoted as $n^u$, i.e., $n^l = \sum_{k=1}^s n_k^l$, and $n^u = \sum_{k=1}^s n_k^u$. The model is evaluated on an unseen target domain $\mathcal{D}_t = \{(\mathbf{x}_i^{(t)}, y_i^{(t)})\}_{i=1}^{n_t}$. $\mathcal{C}^l$ is the label set of labeled data $\mathcal{D}^l = \mathcal{D}_1^l \cup \mathcal{D}_2^l \cup \cdots \cup \mathcal{D}_s^l$, $\mathcal{C}^u$ is the label set of unlabeled data $\mathcal{D}^u = \mathcal{D}_1^u \cup \mathcal{D}_2^u \cup \cdots \cup \mathcal{D}_s^u$, and $\mathcal{C}^t$ is the label set of target domain $\mathcal{D}_t$. Due to the existence of *unknown classes* in unlabeled data and target domain, we have $\mathcal{C}^l \subset \mathcal{C}^u \subset \mathcal{C}^t$. The classes in $\mathcal{C}^l$ are called *known classes*, the classes in $\mathcal{C}^u \backslash \mathcal{C}^l$ are called *seen unknown classes* since they are seen in unlabeled data during the training process and the classes in $\mathcal{C}^t \backslash \mathcal{C}^u$ are called *unseen unknown classes* since they are unseen during the training process. The goal is to learn the model $f^*$ that generalizes well on target domain $\mathcal{D}_t$ for both *known* and *unknown classes*, i.e., for each $(\mathbf{x}_i^{(t)}, y_i^{(t)}) \in \mathcal{D}_t$, if $y_i^{(t)} \in \mathcal{C}^l$, then $f^*(\mathbf{x}_i^{(t)}) = y_i^{(t)}$, else $f^*(\mathbf{x}_i^{(t)}) = unknown$.

To effectively utilize unlabeled data to improve the generalization and robustness of the model, we should carefully exploit *known* and *unknown classes* in them. If known classes are treated as unknown ones, the generalization of the model may decrease significantly; if unknown classes are treated as known ones, the robustness of the model may degrade.

## 3.1 Detecting Known and Unknown Classes

In order to explore unlabeled data and detect known and unknown classes in them, we rely on the outputs of the deep neural networks and the intuition is that known classes will have higher scores than unknown classes [28]. The deep neural network model $f_\Theta = h_\omega \circ g_\theta$ contains two parts, a feature extractor $g_\theta$ and a $|\mathcal{C}^l|$-way linear classifier $h_\omega = [h_\omega^1; \ldots; h_\omega^{|\mathcal{C}^l|}]$. The model maps each sample $\mathbf{x}_i$ into a $|\mathcal{C}^l|$-dimensional logits $\mathbf{z}_i = h_\omega \circ g_\theta(\mathbf{x}_i)$, and then feeds $\mathbf{z}_i$ into a softmax function

---

**Procedure 1** Calculating Thresholds

---

**Input**: Score queues $q^{1:|\mathcal{C}^l|}$, feature extractor $g_\theta$, linear classifiers $h_\omega^{1:|\mathcal{C}^l|}$, validation dataset $\mathcal{D}^v$, number of known classes $|\mathcal{C}^l|$.

**Output**: Temperatures $\tau^{1:|\mathcal{C}^l|}$, thresholds $\delta_{knw}^{1:|\mathcal{C}^l|}$ and $\delta_{unk}^{1:|\mathcal{C}^l|}$.

1: **for** $c = 1$ to $|\mathcal{C}^l|$ **do**
2:     Calculate temperature $\tau^c$ to calibrate $g_\theta$, $h_\omega^c$ on $\mathcal{D}^v$ with Eq.1;
3:     Fit a two-component beta mixture model $\alpha_{1:2}^c$, $\beta_{1:2}^c$ on queue $q^c$ with EM algorithm;
4:     Calculate known and unknown classes thresholds $\delta_{knw}^c = \frac{\alpha_1^c}{\alpha_1^c + \beta_1^c}$, $\delta_{unk}^c = \frac{\alpha_2^c}{\alpha_2^c + \beta_2^c}$.
5: **end for**

---

$\mathcal{S}$ to produce the estimated posterior probability $\hat{\mathbf{p}}_i$ (*a.k.a.* score) of each class, i.e., $\hat{\mathbf{p}}_i = \mathcal{S}\left(f_\Theta(\mathbf{x}_i)\right)$. The training is to minimize the cross-entropy loss $\mathcal{L}_{CE}(\Theta)$ between $\hat{\mathbf{p}}_i$ and one-hot label $\mathbf{y}_i$ over labeled data, i.e., $\Theta^* = \arg\min_\Theta \mathcal{L}_{CE}(\Theta) = \arg\min_\Theta -\frac{1}{n^l}\sum_{i=1}^{n^l}\sum_{c=1}^{|\mathcal{C}^l|}\mathbf{y}_i^c\log\hat{\mathbf{p}}_i^c$, where $n^l$ is the size of labeled data. The gradient of cross-entropy loss w.r.t. linear classifier $h_\omega$ corresponding to class $c$ is $\frac{\partial\mathcal{L}_{CE}(\Theta)}{\partial h_\omega^c} = \frac{1}{n^l}\sum_{i=1}^{n^l}(\hat{\mathbf{p}}_i^c - \mathbf{y}_i^c)g_\theta(\mathbf{x}_i)$. Due to the relative normalization of the softmax function, the score of $\mathbf{x}_i$ on class $c$, i.e., $\hat{\mathbf{p}}_i^c$, not only depends on the logit $\mathbf{z}_i^c$, but also depends on other logits $\mathbf{z}_i^j$, $j \neq c$. Thus, the update of $h_\omega^c$ is affected by the predictions of other classifiers $h_\omega^j$, $j \neq c$. When the logit of $\mathbf{x}_i$ on target class $y_i$ is large, the scores of $\mathbf{x}_i$ on non-target class, i.e., $\hat{\mathbf{p}}_i^c$, $c \neq y_i$, will be small. Since the $\hat{\mathbf{p}}_i^c - \mathbf{y}_i^c$ is small, the gradient of the non-target classifiers may converge to zero. The parameters of the neural network are updated with the stochastic gradient descent (SGD) algorithm, so, the logits of samples on non-target classifiers induced by $h_\omega^j$, $j \neq c$ will not decrease.

In order to update the logits on non-target classifiers induced by $h_\omega^j$, $j \neq c$, we replace the softmax function $\mathcal{S}$ with the sigmoid function $\sigma$. In this way, we obtain an individual score on each way to indicate the probability that one sample belongs to each class, i.e., $\hat{\mathbf{p}}_i^c = \sigma(\mathbf{z}_i^c)$, $c \in \mathcal{C}^l$, and we get $|\mathcal{C}^l|$ one-vs-rest classifiers. With these one-vs-rest classifiers, we can detect known and unknown classes. A straightforward way is to set a fixed threshold $\delta = 0.5$ [39, 38]. Let $j = \arg\max_c(\hat{\mathbf{p}}_i^c)$, if $\hat{\mathbf{p}}_i^j \geq \delta$, it means that it belongs to class $j$; if $\hat{\mathbf{p}}_i^j < \delta$, it means that it is an unknown classes sample. Due to the overconfidence of modern neural networks [40], it is not proper to use such a fixed threshold of 0.5 for all classes. A more reasonable way is to use the class-wise adaptive threshold [41, 42] for each class. To get well-calibrated scores, we first calibrate the classifiers on validation data $\mathcal{D}^v$ with temperature scaling [40]:

$$\tau^c = \arg\min -\sum_{(\mathbf{x}_i, y_i)\in\mathcal{D}^v} \mathbb{I}(y_i = c)\log(\sigma(\mathbf{z}_i^c/\tau^c)) + \mathbb{I}(y_i \neq c)\log(1 - \sigma(\mathbf{z}_i^c/\tau^c)), \quad (1)$$

where $\mathbb{I}(\cdot)$ is an indicator function and $\tau^c$ is the temperature for class $c \in \mathcal{C}^l$. Note that the validation dataset $\mathcal{D}^v$ only contains samples of *known classes* from source domains. And then, we use a two-component beta mixture model to model the score distributions of known classes and unknown classes in an unsupervised way, since it is a more flexible approximator than the Gaussian mixture model [43]. The adaptive thresholds for known classes $\delta_{knw}^j$ (the large one) and unknown classes $\delta_{unk}^j$ (the small one) can be set as the mean values of two fitted beta distributions. Thus, for an unlabeled sample $\mathbf{u}_i \in \mathcal{D}^u$, let $\tilde{\mathbf{p}}_i^j = \sigma(\mathbf{z}_i^j/\tau^j)$ be the scaled score, if $\tilde{\mathbf{p}}_i^j > \delta_{knw}^j$, we consider it belongs to *known class* $j$, i.e. $\hat{y}_i = j$; if $\tilde{\mathbf{p}}_i^j < \delta_{unk}^j$, we consider it belongs to *unknown classes*, i.e. $\hat{y}_i = unknown$; if $\delta_{unk}^j \leq \tilde{\mathbf{p}}_i^j \leq \delta_{knw}^j$, it is difficult to determine its label, hence we set its prediction as *null*, i.e. $\hat{y}_i = null$. Since the model is updated in a mini-batch way, the scores predicted by the model many steps ago are less useful for modeling the score distributions. Hence, we utilize the most recent scores predicted by the model for the fitting of the beta mixture model. In particular, for each class $j$ we maintain a queue $q^j$ with fixed length to record the most recent scaled maximum score of unlabeled sample $\mathbf{u}_i$ if its maximum score belongs to class $j$, i.e., $\arg\max_c(\tilde{\mathbf{p}}_i^c) = j$. The overall procedure is summarized in Procedure 1.

## 3.2 Improving Target Domain Generalization

For labeled training data, we use the following supervised loss,

$$\mathcal{L}^l = -\frac{1}{n^l}\sum_{i=1}^{n^l}\sum_{c=1}^{|\mathcal{C}^l|}\mathbb{I}(y_i = c)\ln(\hat{\mathbf{p}}_i^c) + \frac{1}{|\mathcal{C}^l| - 1}\mathbb{I}(y_i \neq c)\ln(1 - \hat{\mathbf{p}}_i^c). \tag{2}$$

For unlabeled training data, we should exploit them carefully to improve the unseen target domain generalization after having detected known and unknown classes. We construct the weakly and strongly augmented version of each sample following FixMatch [37], and calculate the model's scaled confidence score on its weakly augmented version, i.e., $\tilde{\mathbf{p}}_{i,weak}^c = \sigma\left(f_\Theta\left(T_{weak}(\mathbf{u}_i)\right)/\tau^c\right), c \in \mathcal{C}^l$. Then, we assign pseudo-label $\hat{y}_i$ to it according to Section 3.1, and force the model's prediction on the strongly augmented version $\hat{\mathbf{p}}_{i,strong}^c = \sigma\left(f_\Theta\left(T_{strong}(\mathbf{u}_i)\right)\right), c \in \mathcal{C}^l$ to match the pseudo-label. For unlabeled training data predicted as *known classes* (i.e., $\hat{y}_i \in \mathcal{C}^l$), the loss is defined as

$$\mathcal{L}_{knw}^u = -\frac{1}{n_{knw}^u}\sum_{i=1}^{n_{knw}^u}\sum_{c=1}^{|\mathcal{C}^l|}\mathbb{I}(\hat{y}_i = c)\ln(\hat{\mathbf{p}}_{i,strong}^c) + \frac{1}{|\mathcal{C}^l| - 1}\mathbb{I}(\hat{y}_i \neq c)\ln(1 - \hat{\mathbf{p}}_{i,strong}^c), \tag{3}$$

where $n_{knw}^u$ is the number of samples predicted as *known classes*. For unlabeled training data predicted as *unknown classes* (i.e., $\hat{y}_i = unknown$), the loss is defined as

$$\mathcal{L}_{unk}^u = -\frac{1}{n_{unk}^u}\sum_{i=1}^{n_{unk}^u}\sum_{c=1}^{|\mathcal{C}^l|}\ln(1 - \hat{\mathbf{p}}_{i,strong}^c), \tag{4}$$

where $n_{unk}^u$ is the number of samples predicted as *unknown classes*.

Besides *known* and *unknown classes*, there are some unlabeled samples that are predicted as *null*. In order to fully exploit these data and improve the generalization of the model on the unseen target domain, we adopt Fourier Transformation to disentangle the semantics and styles of the sample [25, 44], and then MixUp [26] the styles of two randomly sampled samples to augment training data for regularizing the model. Specifically, for each $\mathbf{x}$, its Fourier Transformation $\mathcal{F}(\mathbf{x})$ is formulated as:

$$\mathcal{F}(\mathbf{x})(u,v) = \sum_{h=0}^{H-1}\sum_{w=0}^{W-1}\mathbf{x}(h,w)e^{-J2\pi\left(\frac{h}{H}u + \frac{w}{W}v\right)}, \tag{5}$$

where $H, W$ are the height and width of the sample, and $J^2 = -1$. The amplitude component $\mathcal{A}(\mathbf{x})$ and phase component $\mathcal{P}(\mathbf{x})$ are then respectively expressed as

$$\mathcal{A}(\mathbf{x})(u,v) = \left[R^2(\mathbf{x})(u,v) + I^2(\mathbf{x})(u,v)\right]^{1/2}, \mathcal{P}(\mathbf{x})(u,v) = \arctan\left[\frac{I(\mathbf{x})(u,v)}{R(\mathbf{x})(u,v)}\right], \tag{6}$$

where $R(\mathbf{x})$ and $I(\mathbf{x})$ represent the real and imaginary part of $\mathcal{F}(x)$. It is well-known that, the phase component of the Fourier spectrum preserves high-level semantics of the original signal, while the amplitude component contains low-level statistics. Therefore, we could mix up amplitude components of two samples of the same batch while keeping the phase components to generate the augmented samples $\tilde{\mathbf{x}}$ [25, 44], i.e.,

$$\tilde{\mathcal{A}}(\mathbf{x}_i) = (1 - \lambda)\mathcal{A}(\mathbf{x}_i) + \lambda\mathcal{A}(\mathbf{x}_{i'}), \tilde{\mathbf{x}}_i = \mathcal{F}^{-1}\left(\tilde{\mathcal{A}}(\mathbf{x}_i) * e^{-J*\mathcal{P}(\mathbf{x}_i)}\right), \tag{7}$$

where $\mathcal{F}^{-1}(x)$ is the inverse Fourier Transformation, $\lambda \sim U(0,1)$, and $U$ is the uniform distribution.

To avoid overfitting on domain-related low-level statistics, we minimize the consistency regularization loss between the original unlabeled samples and the augmented ones to push the model to pay attention to the high-level semantics of the samples, defined as

$$\mathcal{L}_{con}^u = \frac{1}{n^u}\sum_{i=1}^{n^u}\sum_{c=1}^{|\mathcal{C}^l|}|\hat{\mathbf{p}}_i^c(\mathbf{x}_i) - \hat{\mathbf{p}}_i^c(\tilde{\mathbf{x}}_i)|^2, \tag{8}$$

where $n^u$ is the number of unlabeled data. The overall loss of the training process is formulated as:

$$\mathcal{L} = \mathcal{L}^l + \lambda_1\mathcal{L}_{knw}^u + \lambda_2\mathcal{L}_{unk}^u + \lambda_3\mathcal{L}_{con}^u, \tag{9}$$

where $\lambda_1$, $\lambda_2$ and $\lambda_3$ are hyper-parameters to balance each loss. During the warm-up process, the model is trained only with labeled data, i.e., $\lambda_1 = \lambda_2 = \lambda_3 = 0$. The whole process is summarized in Algorithm 1 (the framework figure can be found in Appendix A of the supplementary material), where the domain index is omitted for brevity.

---

**Algorithm 1** Class-Wise Adaptive Exploration and Exploitation (CWAEE)

---

**Input**: Labeled dataset $\mathcal{D}^l$, unlabeled dataset $\mathcal{D}^u$, validation dataset $\mathcal{D}^v$, training epoch $T$, number of known classes $|\mathcal{C}^l|$.

**Output**: Feature extractor $g_\theta$, linear classifiers $h_\omega^{1:|\mathcal{C}^l|}$.

1: Initialize $g_\theta$, $h_\omega^{1:|\mathcal{C}^l|}$, score queues $q^{1:|\mathcal{C}^l|}$, temperatures $\tau^{1:|\mathcal{C}^l|}$, thresholds $\delta_{knw}^{1:|\mathcal{C}^l|}$, $\delta_{unk}^{1:|\mathcal{C}^l|}$;
2: **for** $t = 1$ to $T$ **do**
3:    **for** $i = 1$ to $max\_iteration$ **do**
4:       Draw a batch of labeled samples $B^l$ and unlabeled samples $B^u$ from $\mathcal{D}^l$ and $\mathcal{D}^u$;
5:       Calculate loss $\mathcal{L}^l$ on $B^l$ with Eq.2;
6:       Predict on $B^u$ with $\tau^{1:|\mathcal{C}^l|}$ to get scaled confidence scores $\tilde{P}^u$;
7:       Split $B^u$ into *known classes* $B_{knw}^u$, *unknown classes* $B_{unk}^u$ and *null* $B_{null}^u$ with $\tilde{P}^u$ and $\delta_{knw}^{1:|\mathcal{C}^l|}$, $\delta_{unk}^{1:|\mathcal{C}^l|}$;
8:       Calculate loss $\mathcal{L}_{knw}^u$ and $\mathcal{L}_{unk}^u$ on $B_{knw}^u$ and $B_{unk}^u$ with Eqs.3 and 4 respectively;
9:       Conduct data augmentation within $B^u$ to get $\tilde{B}^u$ with Eqs.5-7;
10:      Calculate loss $\mathcal{L}_{con}^u$ on $B^u$ and $\tilde{B}^u$ with Eq.8 and get total loss $\mathcal{L}$ with Eq.9;
11:      Backward loss $\mathcal{L}$ and update $g_\theta$, $h_\omega^{1:|\mathcal{C}^l|}$ with SGD;
12:      Update $q^{1:|\mathcal{C}^l|}$ with $\tilde{P}^u$;
13:      Update $\tau^{1:|\mathcal{C}^l|}$, $\delta_{knw}^{1:|\mathcal{C}^l|}$ and $\delta_{unk}^{1:|\mathcal{C}^l|}$ with Procedure 1.
14:    **end for**
15: **end for**

---

## 4 Experiment

### 4.1 Datasets

We use PACS [45], Office-Home [46] and miniDomainNet [47] datasets in the experiments. **PACS** [45] consists of four domains corresponding to four different image styles, including Photo (P), Art painting (A), Cartoon (C) and Sketch (S), and the four domains have the same label set of 7 classes, and contain 9,991 images in total; **OfficeHome** [46] consists of images from four different domains: Artistic (A), Clipart (C), Product (P) and Real-World (R), and has a large domain gap and around 15,500 images of 65 classes; **miniDomainNet** [47] is a subset of DomainNet [48] and has four domains including 18,703 images of Clipart (C), 31,202 images of Painting (P), 65,609 images of Real (R) and 24,492 images of Sketch (S), and it has 126 classes and maintains the complexity of the original DomainNet.

We adopt the common leave-one-domain-out protocol [12, 16]: three domains are used as the source domains and the remaining one as the target domain. Similar to [23, 38] we split the classes into *known* and *unknown classes*, specifically we split the original label set into 3:2:2, 25:20:20 and 42:42:42 (*known classes*, *seen unknown classes* and *unseen unknown classes*) in PACS [45], Office-Home [46] and miniDomainNet [47] respectively in alphabetical order of the class name. On each source domain, 10 labeled samples of each *known class* are randomly sampled to construct the labeled data, and the remaining samples of *known classes* and *seen unknown classes* construct the unlabeled data. All samples on the target domain are used for evaluation.

### 4.2 Compared Methods

We compare our method with various DG methods, OOD detection methods and SSL methods. **DeepAll** naively puts labeled data from all source domains together, and trains the model with Empirical Risk Minimization (ERM); **DAML** [23] conducts meta-learning over augmented domains to learn open-domain generalizable representations for OpenDG problem; **UDG** [32] adopts two classification heads to dynamically separate ID and OOD samples in the unlabeled data and optimizes the model with different learning objectives; **FixMatch** [37] generates highly confident pseudo-labels on weakly-augmented unlabeled samples, and trains the model to predict the pseudo-labels for the strongly-augmented version of the same samples; **OpenMatch**[38] extends FixMatch with OVANet [39] and open-set soft-consistency regularization to detect outliers in unlabeled data for OSSL;

Table 1: Leave-one-domain-out results of *known classes* accuracy (left of the slash) and *unknown classes* AUROC (right of the slash) on PACS, OfficeHome and miniDomainNet.

| PACS | | | | | |
|---|---|---|---|---|---|
| Target Domain | Art | Cartoon | Photo | Sketch | Average |
| DeepAll | 62.96 / 60.06 | 53.41 / 58.15 | 79.17 / 71.26 | 48.60 / 50.16 | 61.03 / 59.91 |
| UDG [32] | 42.98 / 49.83 | 46.92 / 48.52 | 58.75 / 57.28 | 38.82 / 45.21 | 46.87 / 50.21 |
| DAML [23] | 42.07 / 50.27 | 57.74 / 54.80 | 42.87 / 54.00 | 45.29 / 47.20 | 46.99 / 51.57 |
| FixMatch [37] | 81.32 / 68.67 | 61.85 / 56.34 | 85.63 / 64.87 | 76.39 / 48.01 | 76.30 / 59.47 |
| OpenMatch [38] | 83.28 / 68.97 | 75.39 / 66.60 | 91.45 / 68.37 | 58.05 / 47.42 | 77.04 / 62.84 |
| StyleMatch [16] | 82.66 / 63.35 | 71.95 / 56.86 | 90.81 / 67.40 | 77.34 / 43.33 | 80.69 / 57.73 |
| CWAEE | **87.08 / 81.21** | **76.65 / 72.88** | **93.19 / 80.30** | **79.87 / 82.46** | **84.20 / 79.21** |

| OfficeHome | | | | | |
|---|---|---|---|---|---|
| Target Domain | Art | Clipart | Product | Real-World | Average |
| DeepAll | 61.95 / 69.97 | 50.80 / 60.96 | 75.23 / 71.38 | 84.55 / 76.63 | 68.13 / 69.73 |
| UDG [32] | 52.25 / 60.71 | 41.97 / 55.58 | 63.64 / 64.74 | 72.24 / 65.90 | 57.52 / 61.73 |
| DAML [23] | 45.73 / 62.96 | 43.98 / 55.46 | 58.50 / 67.09 | 64.46 / 67.75 | 53.17 / 63.31 |
| FixMatch [37] | 65.25 / 67.60 | 59.32 / 62.18 | 73.31 / 67.72 | 82.35 / 73.16 | 70.06 / 67.67 |
| OpenMatch [38] | 64.95 / 69.27 | 55.82 / 61.60 | 75.20 / 72.93 | 81.76 / 75.71 | 69.43 / 69.90 |
| StyleMatch [16] | 67.83 / 67.40 | 63.02 / 60.15 | 75.46 / 69.16 | 84.79 / 74.44 | 72.77 / 67.79 |
| CWAEE | **70.55 / 75.85** | **64.00 / 66.57** | **76.22 / 76.56** | **86.60 / 81.82** | **74.34 / 75.20** |

| miniDomainNet | | | | | |
|---|---|---|---|---|---|
| Target Domain | Clipart | Painting | Real | Sketch | Average |
| DeepAll | 52.58 / 66.31 | 52.13 / 62.96 | 66.10 / 73.17 | 44.15 / 64.90 | 53.74 / 66.83 |
| UDG [32] | 56.30 / 68.49 | 49.51 / 61.47 | 61.70 / 70.21 | 36.99 / 57.25 | 51.12 / 64.36 |
| DAML [23] | 56.16 / 67.16 | 50.32 / 65.62 | 57.23 / 69.14 | 46.52 / 65.15 | 52.55 / 66.77 |
| FixMatch [37] | 59.71 / 62.83 | 59.71 / 62.37 | 65.63 / 63.58 | 64.78 / 64.90 | 62.01 / 63.42 |
| OpenMatch [38] | 64.53 / 72.70 | 61.55 / 69.80 | **70.61** / 74.87 | 61.40 / 71.30 | 64.52 / 72.17 |
| StyleMatch [16] | 62.42 / 63.63 | 61.23 / 62.21 | 66.02 / 62.58 | 65.44 / 63.46 | 63.77 / 62.97 |
| CWAEE | **66.68 / 73.38** | **65.65 / 73.07** | 69.86 / **75.98** | **66.36 / 74.96** | **67.14 / 74.35** |

**StyleMatch** [16] extends FixMatch with stochastic classifier and multi-view consistency to improve the quality of pseudo-labels and generalization of model respectively for SSDG.

## 4.3 Setup

Most hyper-parameters follow the setting in [16] for a fair comparison. The ImageNet-pretrained ResNet-18 [2] is used as the CNN backbone, and the linear classifiers is implemented with stochastic classifiers. The initial learning rate of SGD optimizer is set to 0.003 for the pretrained backbone and 0.01 for the randomly initialized stochastic classifier, both decaying following the cosine annealing rule. The running epochs are 40, 20 and 20 for PACS, OfficeHome and miniDomainNet respectively. For each mini-batch, we

Table 2: Leave-one-domain-out average AUROC of *seen* (left of the slash) and *unseen unknown classes* (right of the slash) on PACS, OfficeHome and miniDomainNet.

| Dataset | PACS | OfficeHome | miniDomainNet |
|---|---|---|---|
| DeepAll | 58.28 / 60.89 | 68.48 / 71.04 | 69.21 / 67.89 |
| UDG | 50.73 / 49.08 | 63.35 / 60.07 | 64.26 / 64.33 |
| DAML | 50.45 / 52.94 | 62.21 / 64.46 | 67.20 / 66.36 |
| FixMatch | 53.84 / 67.87 | 64.05 / 71.39 | 58.56 / 67.45 |
| OpenMatch | 55.61 / 73.73 | 68.12 / 71.72 | 72.86 / 71.55 |
| StyleMatch | 49.77 / 68.91 | 63.46 / 72.27 | 56.57 / 68.31 |
| CWAEE | **84.09 / 74.53** | **74.57 / 75.87** | **76.81 / 72.31** |

randomly sample 16 labeled samples and 16 unlabeled samples from each source domain. We set $\lambda_1 = 1.0$ on PACS, $\lambda_1 = 0.4$ on OfficeHome and $\lambda_1 = 0.1$ on miniDomainNet. We set $\lambda_2 = 0.4$ and $\lambda_3 = 1.0$ for all three datasets. We evaluate the accuracy on *known classes* and AUROC on *unknown classes* of the methods with 3 different random seeds, and report the average results. More implementation details can be found in Appendix B of the supplementary material.

## 4.4 Results

The results are summarized in Table 1 (the results with standard deviations can be found in Tables 6 and 7 in Appendix C of the supplementary material). From the results, it can be found that our method consistently outperforms the compared methods on all datasets. Although the UDG and DAML methods are developed for *unknown class* detection on the unseen target domain, their performance is much worse than ours. FixMatch has poor classification performance when evaluated on an unseen target domain, due to not considering the domain shifts problem. The performance of our method is much better than that of OpenMatch in most cases since it depends on the fixed thresholds to detect *unknown classes* in unlabeled data. StyleMatch has a worse performance than ours since it treats all unlabeled samples as one of the known classes. In order to evaluate the detection performance of methods on *seen* and *unseen unknown classes* respectively, we also report the AUROC of *seen* and *unseen unknown classes* on the target domain in Table 2 (the results with standard deviations can be found in Table 8 in Appendix C of the supplementary material). It can be found that our method has better detection performance on both *seen* and *unseen unknown classes*.

## 4.5 Ablation Study

In order to provide additional insight into what makes our method successful, we conduct ablation experiments on OfficeHome.

### 4.5.1 Effectiveness of Used Modules

Our method contains several modules, including supervised loss (Eq.2), unsupervised loss on *known classes* (Eq.3), unsupervised loss on *unknown classes* (Eq.4), class-wise adaptive thresholds (Procedure 1) and consistency regularization loss (Eq.8). The results with standard deviations are presented in Table 3. With the supervised loss, our

Table 3: Ablation study on the used modules.

| Ablation | Accuracy / AUROC |
|---|---|
| Supervised loss | 68.13±0.88 / 69.73±1.01 |
| + Unsupervised loss on known classes with fixed thresholds | 72.30±0.18 / 64.47±0.60 |
| + Unsupervised loss on unknown classes with fixed thresholds | 72.38±0.16 / 65.16±0.43 |
| + Class-wise adaptive thresholds | 73.70±0.39 / 74.44±0.92 |
| + Consistency regularization loss | **74.34±0.35 / 75.20±0.74** |

method trains the model only with labeled samples. After detecting known and unknown classes with the fixed thresholds (i.e., $\delta_{knw}^{1:|\mathcal{C}^l|} = \delta_{unk}^{1:|\mathcal{C}^l|} = 0.5$) and trained with the unsupervised loss on *known* and *unknown classes*, the performance can be improved with these detected samples. By replacing the fixed thresholds with class-wise adaptive thresholds, the performance can be further boosted. With all the used modules, we can get the best performance.

To better show the effectiveness of the class-wise adaptive thresholds, we compare the quality of pseudo-labels when trained with fixed thresholds and the class-wise adaptive thresholds. The results are depicted in Figure 2. It can be found that the accuracy of pseudo-labels assigned by our method is higher than that of compared method on *known classes* and *unknown classes* in all 4 domains, which means that our method has better performance on *known classes* classification and *unknown classes* detection.

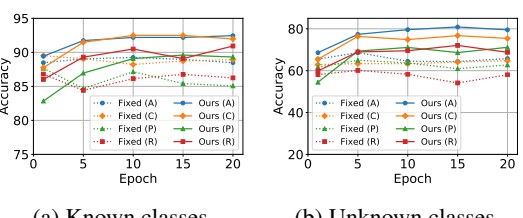

(a) Known classes     (b) Unknown classes

Figure 2: The accuracy of pseudo-labels on unlabeled samples.

### 4.5.2 Sensitivity to Number of Seen Unknown Classes

In order to study the sensitivity of our method to the number of *seen unknown classes*, we compare our method with existing SSL methods under different numbers of *seen unknown classes* ($|\mathcal{C}^u \backslash \mathcal{C}^l|$ varies from 0 to 30), and the results are depicted in Figure 3. It can be found that our

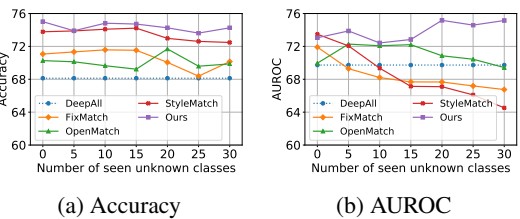

(a) Accuracy     (b) AUROC

Figure 3: The results with various numbers of seen unknown classes.

method outperforms the compared methods. With the increase in the number of *seen unknown classes*, the AUROC on *unknown classes* obtained by our method increases while the accuracy on *known classes* is maintained.

### 4.5.3 Sensitivity to Number of Labeled Samples

In order to study the sensitivity of our method to the number of labeled samples, we compare our method with existing methods under different numbers of labeled samples (5, 10 and 20 labels of each *known class* on each domain) and the results are summarized in Table 4 (the results with standard deviations can be found in Table 9 in Appendix C of the supplementary material). It can be found that our method outperforms compared methods under different numbers of labeled samples. The performance of our method under extremely few labels (i.e., 5 labels of each class on each domain) is promising, since it can exploit *known* and *unknown classes* in unlabeled data in a more reasonable way.

Table 4: Average accuracy / AUROC on OfficeHome with different numbers of labeled samples.

| # Labels | 5 | 10 | 20 |
|---|---|---|---|
| DeepAll | 62.87 / 67.77 | 68.13 / 69.73 | 72.17 / 70.83 |
| UDG | 47.02 / 55.68 | 57.52 / 61.73 | 65.31 / 67.66 |
| DAML | 52.67 / 64.15 | 53.17 / 63.31 | 50.95 / 63.07 |
| FixMatch | 66.90 / 64.29 | 70.06 / 67.67 | 71.92 / 70.21 |
| OpenMatch | 66.24 / 69.81 | 69.43 / 69.90 | 70.14 / 69.81 |
| StyleMatch | 70.16 / 63.57 | 72.77 / 67.79 | 75.36 / 70.45 |
| CWAEE | **71.24 / 73.23** | **74.34 / 75.20** | **76.09 / 73.96** |

### 4.5.4 Histograms of Confidence Scores

The empirical probability density function of confidence scores of unlabeled data in different queues and different epochs are depicted in Figure 4. Here, we report class 0 and class 1 in different epochs as examples, i.e., Figure 4a vs 4b, Figure 4c vs 4d, which shows that the confidence scores have different distributions in the different training epochs. It can be found that our method CWAEE can find the adaptive thresholds for *known* and *unknown classes*.

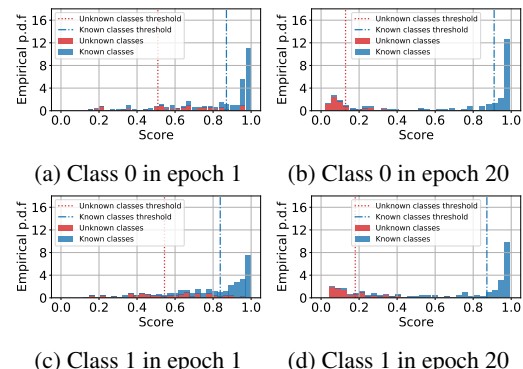

(a) Class 0 in epoch 1    (b) Class 0 in epoch 20

(c) Class 1 in epoch 1    (d) Class 1 in epoch 20

Figure 4: The empirical p.d.f. of the confidence scores of unlabeled data.

## 5 Conclusion

In this paper, we focus on a more realistic semi-supervised domain generalization scenario, where *known classes* and *seen unknown classes* are mixed in unlabeled training data while *known classes*, *seen unknown classes* and *unseen unknown classes* are mixed in testing data. In order to utilize unlabeled training data, we detect known and unknown classes in them with the class-wise adaptive thresholds based on one-vs-rest classifiers. We adopt consistency regularization on augmented training samples based on Fourier Transformation to improve the generalization on the unseen target domain. The experimental results conducted on different real-world datasets show that our method outperforms the existing state-of-the-art methods.

### Broader Impacts

Our work provides an effective method for semi-supervised domain generalization with *known* and *unknown classes*. We believe our work will be beneficial for domain generalization projects with few labeled data and many unlabeled data, and does not have negative societal impacts.

## Acknowledgments

This work is supported by the National Science Foundation of China (62276125, 61921006), the Fundamental Research Funds for the Central Universities (022114380013), and the Collaborative Innovation Center of Novel Software Technology and Industrialization.

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
