# A    Framework of CWAEE

For a better understanding of our method, we give the framework of CWAEE. We use the outputs of the one-vs-rest classifiers to detect *known* and *unknown classes* in unlabeled data. Each unlabeled sample $\mathbf{u}_i \in \mathcal{D}^u$ is passed through the network and gets its calibrated score $\tilde{\mathbf{p}}_i^c, c \in \mathcal{C}^l$ on each one-vs-rest classifier, which indicates the probability that it belongs to this class. Each sample and its scores are appended to corresponding queues according to the maximum score of all $|\mathcal{C}^l|$ scores, i.e. sample $\mathbf{u}_i$ is appended to $q^j$, where $j = \operatorname{argmax}_c (\tilde{\mathbf{p}}_i^c)$, and $|\mathcal{C}^l|$ is the number of *known classes*. Then, the class-wise adaptive threshold is calculated with a two-component beta mixture model (BMM) which models the score distributions of *known classes* and *unknown classes* in an unsupervised way. The adaptive thresholds for known classes $\delta_{knw}^j$ (the large one) and unknown classes $\delta_{unk}^j$ (the small one) are set as the mean values of two fitted beta distributions. The unlabeled sample $\mathbf{u}_i$ can be determined whether belongs to a *known class*, an *unknown class*, or *null* according to Section 3.1 in the main text. The entire process is summarized in Figure 5.

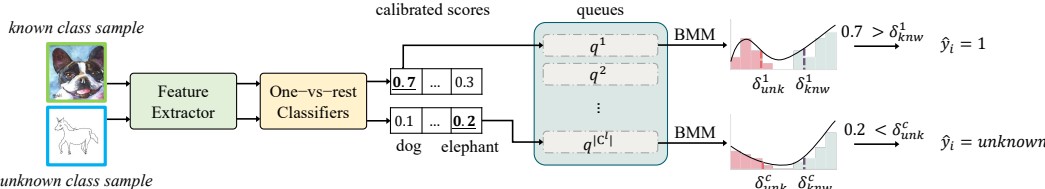

Figure 5: The process of detecting *known* and *unknown classes*.

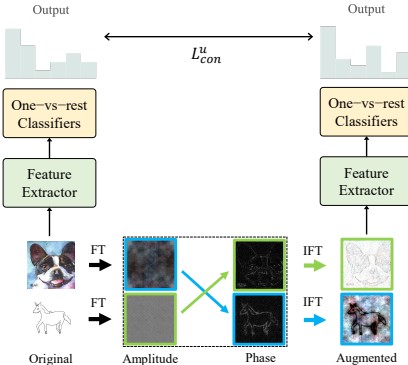

Figure 6: The process of improving target domain generalization.

For Domain Generalization, it is important to exploit the inter-domain information which includes domain-dependent styles and domain-invariant semantics. The amplitude component of the Fourier spectrum contains the styles of the samples, while the phase component of the Fourier spectrum preserves the semantics of the samples [1, 2], so we can disentangle styles and semantics with Fourier Transform (FT) and get augmented samples with amplitude component mixing and Inverse Fourier Transform (IFT) to enrich the inter-domain information. The consistency regularization loss between the original unlabeled samples and the augmented ones is minimized to push the model to pay attention to the high-level semantics of the samples. The process is summarized in Figure 6.

# B    Details of the Experiment

## B.1    Datasets

We conduct experiments on PACS [3], OfficeHome [4] and miniDomainNet [5] datasets. For each dataset, we split the original label set into *known classes*, *seen unknown classes* and *unseen unknown classes* in alphabetical order of the class name, as shown in Table 5.

Table 5: The splits of used datasets.

| Dataset | Known Classes | Seen Unknown Classes | Unseen Unknown Classes |
|---|---|---|---|
| PACS [3] | Dog, Elephant, Giraffe | Guitar, Horse | House, Person |
| OfficeHome [4] | Alarm Clock, Backpack, Batteries, Bed, Bike, Bottle, Bucket, Calculator, Calendar, Candles, Chair, Clipboards, Computer, Couch, Curtains, Desk Lamp, Drill, Eraser, Exit Sign, Fan, File Cabinet, Flipflops, Flowers, Folder, Fork | Glasses, Hammer, Helmet, Kettle, Keyboard, Knives, Lamp Shade, Laptop, Marker, Monitor, Mop, Mouse, Mug, Notebook, Oven, Pan, Paper Clip, Pen, Pencil, Postit Notes | Printer, Push Pin, Radio, Refrigerator, Ruler, Scissors, Screwdriver, Shelf, Sink, Sneakers, Soda, Speaker, Spoon, Table, Telephone, ToothBrush, Toys, Trash Can, TV, Webcam |
| miniDomainNet [5] | Aircraft Carrier, Alarm Clock, Ant, Anvil, Asparagus, Axe, Banana, Basket, Bathtub, Bear, Bee, Bird, Blackberry, Blueberry, Bottlecap, Broccoli, Bus, Butterfly, Cactus, Cake, Calculator, Camel, Camera, Candle, Cannon, Canoe, Carrot, Castle, Cat, Ceiling Fan, Cell Phone, Cello, Chair, Chandelier, Coffee Cup, Compass, Computer, Cow, Crab, Crocodile, Cruise Ship, Dog | Dolphin, Dragon, Drums, Duck, Dumbbell, Elephant, Eyeglasses, Feather, Fence, Fish, Flamingo, Flower, Foot, Fork, Frog, Giraffe, Goatee, Grapes, Guitar, Hammer, Helicopter, Helmet, Horse, Kangaroo, Lantern, Laptop, Leaf, Lion, Lipstick, Lobster, Microphone, Monkey, Mosquito, Mouse, Mug, Mushroom, Onion, Panda, Peanut, Pear, Peas, Pencil | Penguin, Pig, Pillow, Pineapple, Potato, Power Outlet, Purse, Rabbit, Raccoon, Rhinoceros, Rifle, Saxophone, Screwdriver, Sea Turtle, See Saw, Sheep, Shoe, Skateboard, Snake, Speedboat, Spider, Squirrel, Strawberry, Streetlight, String Bean, Submarine, Swan, Table, Teapot, Teddy-Bear, Television, The Eiffel Tower, The Great Wall Of China, Tiger, Toe, Train, Truck, Umbrella, Vase, Watermelon, Whale, Zebra |

We resize each image to 224*224. The weak augmentation includes random flip and random translation, while the strong augmentation includes random flip, RandAugment [6], Cutout [7] and AdaIN [8].

## B.2 Implementation Details

**Hyper-parameters.** Unless otherwise noted, the same value is used for all datasets.

- The CNN backbone, ImageNet-pretrained ResNet-18 [9].
- The linear classifier, stochastic classifier [10].
- The batch-size is set to 16 per source domain, both for labeled data and unlabeled data.
- The learning rate is set to 0.003 for the pretrained backbone and 0.01 for the randomly initialized linear classifier. They decay following the cosine annealing rule.
- The optimizer is SGD with momentum = 0.9, and weight_decay = 5e-4.
- The running epochs are 40, 20 and 20 (counted by unlabeled dataset) for PACS, OfficeHome and miniDomainNet respectively.
- The warm-up epoch is 5 (counted by labeled dataset).
- The trade-off parameter $\lambda_1 = 1.0$ on PACS, $\lambda_1 = 0.4$ on OfficeHome and $\lambda_1 = 0.1$ on miniDomainNet.
- Trade-off parameters $\lambda_2 = 0.4$, $\lambda_3 = 1.0$.
- The queue length is set to 300 for each *known class*.

**Train-Validate-Test Split.** We use the official train-validate split of each dataset for validation, and labeled samples are randomly sampled from the training split. All samples of the target domain are used as test data.

We run all experiments on 8 Nvidia A6000 GPUs.

## C Additional Experimental Results

### C.1 Sensitivity to Hyper-parameters

We test the sensitivity of hyper-parameters $\lambda_1$, $\lambda_2$, and $\lambda_3$ on OfficeHome. For each hyper-parameter, we vary it from 0 to 1 while keeping others fixed with default values. The results are shown in Figure 7. With the increase of $\lambda_1$, more unlabeled samples will be detected as known classes. At first,

samples belonging to known classes will be selected correctly, then more and more unknown class samples will be selected, thus the accuracy and AUROC will first increase and then decrease. With the increase of $\lambda_2$, more unlabeled samples will be detected as unknown classes, thus the accuracy will be harmed when known class samples are detected as unknown classes. With the increase of $\lambda_3$, both the accuracy and AUROC increase to saturation. In summary, the performance is stable near the default values of the hyper-parameters, and will degrade if any loss term is not employed (i.e. weight = 0). We recommend setting $\lambda_1 = 1.0$, $\lambda_2 = 0.4$, and $\lambda_3 = 1.0$ as the starting point of hyper-parameters searching.

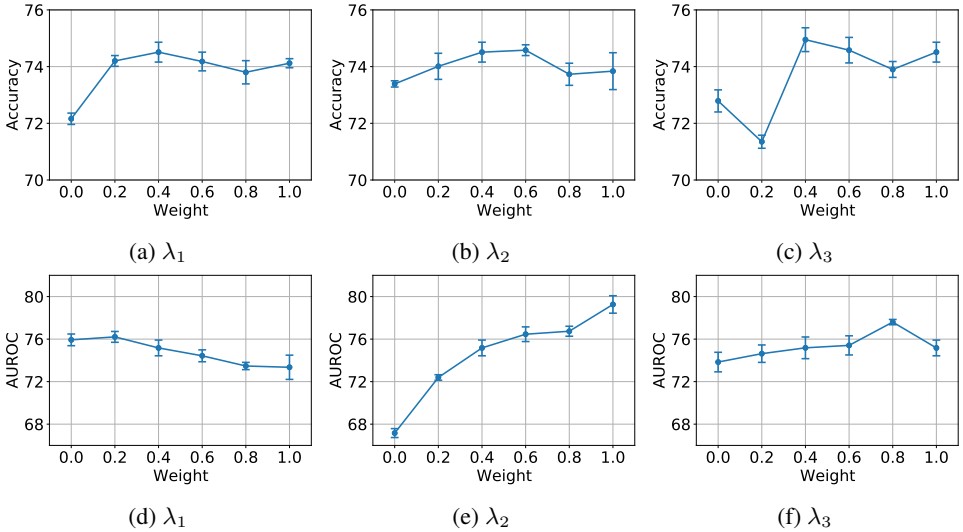

Figure 7: The accuracy (a, b, c) and AUROC (d, e, f) sensitivity with respect to the hyper-parameters $\lambda_1$, $\lambda_2$, $\lambda_3$ on OfficeHome.

## C.2 Results with Standard Deviations

The results with standard deviations of Table 1 in the main text are summarized in Tables 6 and 7 respectively. The results with standard deviations of Table 2 in the main text are summarized in Table 8. The results with standard deviations of Table 4 in the main text are summarized in Table 9.

## C.3 Results without Pre-trained Model

It is a common practice to use ImageNet-pretrained model as the backbone network and train the model for a few epochs on the training datasets in Domain Generalization [12, 1, 10]. We follow the training setting of the seminal work [10] to use the ImageNet-pretrained model . In order to study the effects of the pretrained model on the performance of the proposed method, we train the model from scratch with the compared methods and ours. The results on OfficeHome with 25:20:20 (*known classes*, *seen unknown classes* and *unseen unknown classes*) are summarized in Tables 10 and 11. It can be found that our method outperforms compared methods at most cases when the model is trained from scratch. Although the *unknown classes* AUROC of OpenMatch [14] is better than ours on Clipart, our method significantly outperforms OpenMatch [14] on other domains (Art, Product and Real-World).

## C.4 Results on Additional Benchmarks

In order to verify the effectiveness of the proposed method on more challenging datasets, we conduct the experiments on the FMoW dataset from the WILDS benchmark [15]. It is a satellite image classification task that has 62 classes and 80 domains (16 years $\times$ 5 regions). In particular, the input $x$ is a RGB satellite image of size $224 \times 224$, the label $y$ is one of the 62 building or land use categories, and the domain $d$ represents the year that the image was taken as well as its corresponding geographical region (Africa, the Americas, Oceania, Asia, or Europe). The train/test/validation splits

Table 6: Leave-one-domain-out results of *known classes* accuracy on PACS, OfficeHome and miniDomainNet.

| | PACS | | | | |
|---|---|---|---|---|---|
| Target Domain | Art | Cartoon | Photo | Sketch | Average |
| DeepAll | 62.96±2.28 | 53.41±9.79 | 79.17±8.28 | 48.60±6.33 | 61.03±3.54 |
| UDG [11] | 42.98±4.43 | 46.92±7.55 | 58.75±8.55 | 38.82±8.76 | 46.87±4.22 |
| DAML [12] | 42.07±6.54 | 57.74±6.95 | 42.87±4.93 | 45.29±0.90 | 46.99±4.05 |
| FixMatch [13] | 81.32±4.18 | 61.85±4.96 | 85.63±7.21 | 76.39±4.09 | 76.30±0.76 |
| OpenMatch [14] | 83.28±1.73 | 75.39±4.15 | 91.45±2.93 | 58.05±11.33 | 77.04±1.75 |
| StyleMatch [10] | 82.66±3.44 | 71.95±3.74 | 90.81±2.24 | 77.34±5.39 | 80.69±0.77 |
| CWAEE | **87.08±0.06** | **76.65±6.46** | **93.19±1.55** | **79.87±2.28** | **84.20±2.01** |

| | OfficeHome | | | | |
|---|---|---|---|---|---|
| Target Domain | Art | Clipart | Product | Real-World | Average |
| DeepAll | 61.95±0.41 | 50.80±1.58 | 75.23±0.79 | 84.55±1.73 | 68.13±0.88 |
| UDG [11] | 52.25±2.98 | 41.97±2.13 | 63.64±1.50 | 72.24±1.67 | 57.52±1.22 |
| DAML [12] | 45.73±2.62 | 43.98±6.43 | 58.50±3.88 | 64.46±6.39 | 53.17±0.71 |
| FixMatch [13] | 65.25±1.89 | 59.32±3.00 | 73.31±1.58 | 82.35±0.14 | 70.06±1.22 |
| OpenMatch [14] | 64.95±0.37 | 55.82±0.31 | 75.20±0.25 | 81.76±0.50 | 69.43±0.18 |
| StyleMatch [10] | 67.83±0.55 | 63.02±1.50 | 75.46±0.40 | 84.79±0.51 | 72.77±0.58 |
| CWAEE | **70.55±0.92** | **64.00±0.72** | **76.22±1.05** | **86.60±0.96** | **74.34±0.35** |

| | miniDomainNet | | | | |
|---|---|---|---|---|---|
| Target Domain | Clipart | Painting | Real | Sketch | Average |
| DeepAll | 52.58±1.52 | 52.13±0.71 | 66.10±1.88 | 44.15±2.27 | 53.74±1.23 |
| UDG [11] | 56.30±1.44 | 49.51±1.81 | 61.70±2.63 | 36.99±1.40 | 51.12±1.37 |
| DAML [12] | 56.16±4.18 | 50.32±0.89 | 57.23±1.05 | 46.52±3.60 | 52.55±1.51 |
| FixMatch [13] | 57.91±0.71 | 59.71±0.50 | 65.63±0.80 | 64.78±1.14 | 62.01±0.64 |
| OpenMatch [14] | 64.53±1.41 | 61.55±1.00 | **70.61±0.87** | 61.40±1.29 | 64.52±0.38 |
| StyleMatch [10] | 62.42±1.91 | 61.23±0.36 | 66.02±0.93 | 65.44±0.65 | 63.77±0.60 |
| CWAEE | **66.68±0.98** | **65.65±1.20** | 69.86±0.22 | **66.36±1.17** | **67.14±0.17** |

are based on the time when the images are taken. Specifically, images taken before 2013 are used as the training set. Images taken between 2013 and 2015 are used as the validation set. Images taken after 2015 are used for testing. We split the original label set into 42:20:20 (*known classes*, *seen unknown classes* and *unseen unknown classes*) in alphabetical order of the class name. 15 labeled samples of each *known class* are randomly sampled to construct the labeled data, and the remaining samples of *known classes* and *seen unknown classes* construct the unlabeled data. We follow the setup of [16] which uses ImageNet-pretrained DenseNet121 [17] and Adam [18] optimizer, sets learning rate as 1e-4, weight_decay as 0, batch-size as 32, and trains 5 epochs. We set $\lambda_1 = 0.2$, $\lambda_2 = 0.4$, $\lambda_3 = 0.25$ in our method. We compare our method with existing methods except DAML [12] which is out of memory since it requires to train one model on each domain. The results are summarized in Table 12. It can be found that our method outperforms compared methods in this benchmark.

## C.5 Training Epochs

We follow the training setting of the seminal work [10] of SSDG for a fair comparison, which sets running epochs as 40, 20 and 20 for PACS, OfficeHome and miniDomainNet respectively. In order to study the effects of training epochs on the performance of the proposed method and compared methods. We extend the training epochs to 40 on OfficeHome and conduct the experiments. The results are shown in Figure 8. It can be found that the performance of all methods does not change much after 20 epochs, and our method keeps outperforming others.

Table 7: Leave-one-domain-out results of *unknown classes* AUROC on PACS, OfficeHome and miniDomainNet.

| Target Domain | Art | Cartoon | Photo | Sketch | Average |
|---|---|---|---|---|---|
| | *PACS* | | | | |
| DeepAll | 60.06±2.06 | 58.15±1.33 | 71.26±11.47 | 50.16±2.98 | 59.91±4.23 |
| UDG [11] | 49.83±3.46 | 48.52±4.71 | 57.28±9.62 | 45.21±2.41 | 50.21±2.26 |
| DAML [12] | 50.27±3.81 | 54.80±2.25 | 54.00±5.63 | 47.20±6.37 | 51.57±1.34 |
| FixMatch [13] | 68.67±2.68 | 56.34±3.82 | 64.87±5.40 | 48.01±3.66 | 59.47±0.76 |
| OpenMatch [14] | 68.97±3.27 | 66.60±5.89 | 68.37±6.35 | 47.42±6.75 | 62.84±4.01 |
| StyleMatch [10] | 63.35±3.65 | 56.86±3.12 | 67.40±4.31 | 43.33±0.61 | 57.73±2.61 |
| CWAEE | **81.21±1.59** | **72.88±4.01** | **80.30±3.81** | **82.46±1.35** | **79.21±1.33** |

| Target Domain | Art | Clipart | Product | Real-World | Average |
|---|---|---|---|---|---|
| | *OfficeHome* | | | | |
| DeepAll | 69.97±0.30 | 60.96±1.04 | 71.38±1.47 | 76.63±2.04 | 69.73±1.01 |
| UDG [11] | 60.71±1.40 | 55.58±1.94 | 64.74±0.40 | 65.90±0.96 | 61.73±0.42 |
| DAML [12] | 62.96±3.40 | 55.46±2.63 | 67.09±1.69 | 67.75±3.12 | 63.31±0.46 |
| FixMatch [13] | 67.60±0.61 | 62.18±0.19 | 67.72±0.33 | 73.16±0.52 | 67.67±0.21 |
| OpenMatch [14] | 69.27±0.45 | 61.60±0.92 | 72.93±1.68 | 75.71±0.90 | 69.90±0.34 |
| StyleMatch [10] | 67.40±0.54 | 60.15±1.51 | 69.16±1.63 | 74.44±1.06 | 67.79±0.86 |
| CWAEE | **75.85±0.99** | **66.57±2.98** | **76.56±1.97** | **81.82±1.94** | **75.20±0.74** |

| Target Domain | Clipart | Painting | Real | Sketch | Average |
|---|---|---|---|---|---|
| | *miniDomainNet* | | | | |
| DeepAll | 66.31±2.09 | 62.96±0.20 | 73.17±0.88 | 64.90±0.34 | 66.83±0.56 |
| UDG [11] | 68.49±1.01 | 61.47±1.03 | 70.21±0.50 | 57.25±0.49 | 64.36±0.18 |
| DAML [12] | 67.16±1.86 | 65.62±0.21 | 69.14±0.28 | 65.15±2.26 | 66.77±0.87 |
| FixMatch [13] | 62.83±0.99 | 62.37±0.83 | 63.58±0.31 | 64.90±0.38 | 63.42±0.57 |
| OpenMatch [14] | 72.70±2.23 | 69.80±0.75 | 74.87±0.25 | 71.30±0.62 | 72.17±0.72 |
| StyleMatch [10] | 63.63±1.30 | 62.21±0.78 | 62.58±0.12 | 63.46±0.97 | 62.97±0.31 |
| CWAEE | **73.38±0.71** | **73.07±0.95** | **75.98±0.92** | **74.96±0.52** | **74.35±0.43** |

Table 8: Leave-one-domain-out average AUROC of *seen* (left of the slash) and *unseen unknown classes* (right of the slash) on PACS, OfficeHome and miniDomainNet.

| Dataset | PACS | OfficeHome | miniDomainNet |
|---|---|---|---|
| DeepAll | 58.28±5.18 / 60.89±4.73 | 68.48±1.58 / 71.04±0.47 | 69.21±0.58 / 67.89±0.78 |
| UDG [11] | 50.73±0.37 / 49.08±3.30 | 63.35±0.71 / 60.07±0.49 | 64.26±0.37 / 64.33±0.21 |
| DAML [12] | 50.45±1.14 / 52.94±7.19 | 62.21±0.67 / 64.46±0.81 | 67.20±1.05 / 66.36±0.76 |
| FixMatch [13] | 53.84±1.79 / 67.87±1.62 | 64.05±0.61 / 71.39±0.38 | 58.56±0.71 / 67.45±0.49 |
| OpenMatch [14] | 55.61±6.70 / 73.73±1.52 | 68.12±0.91 / 71.72±0.19 | 72.86±0.90 / 71.55±0.61 |
| StyleMatch [10] | 49.77±2.54 / 68.91±4.28 | 63.46±1.34 / 72.27±0.38 | 56.57±0.64 / 68.31±0.06 |
| CWAEE | **84.09±0.43 / 74.53±2.29** | **74.57±1.06 / 75.87±0.55** | **76.81±0.30 / 72.31±0.53** |

Table 9: Leave-one-domain-out average *known classes* accuracy (left of the slash) and *unknown classes* AUROC (right of the slash) on OfficeHome with different numbers of labeled samples.

| # Labels | 5 | 10 | 20 |
|---|---|---|---|
| DeepAll | 62.87±1.21 / 67.77±1.39 | 68.13±0.88 / 69.73±1.01 | 72.17±0.57 / 70.83±0.45 |
| UDG [11] | 47.02±0.90 / 55.68±0.51 | 57.52±1.22 / 61.73±0.42 | 65.31±0.50 / 67.66±0.29 |
| DAML [12] | 52.67±0.68 / 64.15±0.99 | 53.17±0.71 / 63.31±0.46 | 50.95±2.53 / 63.07±0.33 |
| FixMatch [13] | 66.90±0.75 / 64.29±0.85 | 70.06±1.22 / 67.67±0.21 | 71.92±0.61 / 70.21±0.73 |
| OpenMatch [14] | 66.24±0.99 / 69.81±0.20 | 69.43±0.18 / 69.90±0.34 | 70.14±0.73 / 69.81±0.41 |
| StyleMatch [10] | 70.16±0.53 / 63.57±1.46 | 72.77±0.58 / 67.79±0.86 | 75.36±0.35 / 70.45±0.21 |
| CWAEE | **71.24±0.43 / 73.23±0.60** | **74.34±0.35 / 75.20±0.74** | **76.09±0.36 / 73.96±0.57** |

Table 10: Leave-one-domain-out results of *known classes* accuracy on OfficeHome without pretrained model.

| Target Domain | Art | Clipart | Product | Real-World | Average |
|---|---|---|---|---|---|
| DeepAll | 17.17±1.35 | 16.47±1.73 | 25.51±1.93 | 26.06±1.82 | 21.30±1.19 |
| UDG [11] | 16.25±3.50 | 13.41±2.16 | 24.20±0.73 | 23.34±2.40 | 19.30±1.51 |
| DAML [12] | 25.31±1.52 | 25.25±0.54 | 37.23±3.98 | 35.28±2.34 | 30.77±1.68 |
| FixMatch [13] | 34.28±1.47 | 33.89±3.18 | 44.22±2.32 | 46.16±2.81 | 39.63±1.30 |
| OpenMatch [14] | 26.32±1.49 | 22.80±2.91 | 32.73±0.89 | 34.74±0.92 | 29.15±0.91 |
| StyleMatch [10] | 37.00±0.96 | **35.76±1.70** | 45.08±0.76 | 48.22±1.37 | 41.51±1.03 |
| CWAEE | **37.83±0.47** | 35.58±1.86 | **46.61±2.10** | **51.64±1.99** | **42.91±0.79** |

Table 11: Leave-one-domain-out results of *unknown classes* AUROC on OfficeHome without pretrained model.

| Target Domain | Art | Clipart | Product | Real-World | Average |
|---|---|---|---|---|---|
| DeepAll | 52.62±1.15 | 50.88±0.12 | 56.63±1.01 | 54.61±0.82 | 53.68±0.55 |
| UDG [11] | 50.83±1.92 | 52.74±2.32 | 59.67±1.24 | 54.03±2.81 | 54.32±1.99 |
| DAML [12] | 55.31±0.46 | 51.81±0.61 | 59.57±2.43 | 58.14±1.57 | 56.21±1.07 |
| FixMatch [13] | 57.08±0.51 | 53.57±0.76 | 60.80±0.97 | 57.43±1.10 | 57.22±0.28 |
| OpenMatch [14] | 57.00±2.37 | **54.10±0.43** | 58.52±2.05 | 57.61±0.32 | 56.81±0.51 |
| StyleMatch [10] | 57.74±1.79 | 50.98±1.94 | 58.51±1.75 | 57.63±1.20 | 56.22±0.43 |
| CWAEE | **65.17±1.08** | 49.67±1.28 | **65.92±0.35** | **61.40±1.55** | **60.54±0.68** |

Table 12: Results of *known classes* accuracy and *unknown classes* AUROC on FMoW.

| | Accuracy | AUROC |
|---|---|---|
| DeepAll | 17.36±1.89 | 50.08±0.84 |
| UDG [11] | 13.58±1.59 | 51.10±1.38 |
| FixMatch [13] | 18.22±1.69 | 51.15±2.06 |
| OpenMatch [14] | 6.55±3.87 | 49.58±1.66 |
| StyleMatch [10] | 17.61±6.39 | 49.75±2.07 |
| CWAEE | **19.43±4.97** | **52.58±2.67** |

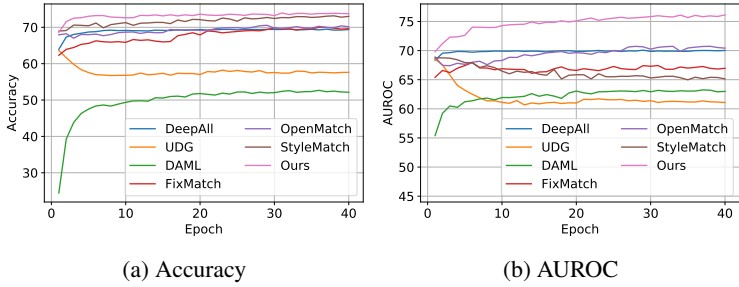

(a) Accuracy      (b) AUROC

Figure 8: Leave-one-domain-out average *known classes* accuracy (a) and *unknown classes* AUROC (b) with respect to the training epochs on OfficeHome.