# OpenReview forum: "Semi-Supervised Domain Generalization with Known and Unknown Classes"
_NeurIPS.cc/2023/Conference — NeurIPS 2023 spotlight_

### Official Review · Reviewer_kd1R · 2023-07-04

**Soundness:** 3 good
**Presentation:** 3 good
**Contribution:** 3 good
**Rating:** 5
**Confidence:** 3

**Summary:**

This paper introduces a new methodology for Semi-Supervised Domain Generalization (SSDG). This paper proposes the Class-Wise Adaptive Exploration and Exploitation (CWAEE) method, which contains one-vs-rest classifiers, class-wise adaptive thresholds, and consistency regularization based on Fourier Transformation. This algorithm shows the improvement in their SSDG settings.

**Strengths:**

Pros.
- This paper is well-written, motivated, and easy to follow.
- This paper proposes several methods in which class-wise adaptive thresholds and consistency regularization based on Fourier Transformation are very interesting works. (If the overview image for consistency regularization in the appendix is added in the main paper, it would be much easy to understand the concept. It is very novel and interesting. )
- This paper shows extensive experiments and shows the effectiveness of this method.

**Weaknesses:**

Cons.
- (Major) This paper didn’t follow the previous training pipeline for the baseline. In FixMatch, it trains the model 2^20 iterations. For Cifar10/Cifar100, it is about 10k epochs. With only 80 epochs, I would say it is not the original other baseline numbers.
- (Major) This paper uses Imagenet pre-trained model. It means that unlabeled data could be trained in the pre-training phase.
- (Major) This paper evaluates their algorithm in only their setting. I think it can be evaluated in the same OOD setting in OpenMatch. (known/unknown in cifar10/cifar100 or Imagenet) If this paper could show the improvement in this setting, it would be more solid results.
- (Minor) The one-vs-rest classifier is very similar to One-vs-All Outlier Detector in OpenMatch. (Novelty)

**Questions:**

Please see the weaknesses.

**Limitations:**

Please see the weaknesses.

---

> ### Author Rebuttal · Authors · 2023-08-09
>
> W1: (Major) This paper didn’t follow the previous training pipeline for the baseline. In FixMatch, it trains the model 2\^20 iterations. For Cifar10/Cifar100, it is about 10k epochs. With only 80 epochs, I would say it is not the original other baseline numbers.
>
> A: Our paper focuses on the Semi-Supervised Domain Generalization (SSDG), thus we follow the training setting of the seminal work [11] of SSDG for a fair comparison.
>
> [11] Zhou et al, Semi-Supervised Domain Generalization with Stochastic StyleMatch, NeurIPS 2021 Workshop (extension version published in IJCV'23).
>
> W2: (Major) This paper uses Imagenet pre-trained model. It means that unlabeled data could be trained in the pre-training phase.
>
> A: We follow the training setting of the seminal work [11] to use the ImageNet pre-trained model.
>
> The results of the proposed method trained from scratch on Officehome with 25:20:20 (*known classes*, *seen unknown classes* and *unseen unknown classes*) are summarized in Table 2. It can be found that our method outperforms other compared methods when the model is trained from scratch.
>
> Table 2: Leave-one-domain-out average results of *known classes* accuracy and *unknown classes* AUROC on OfficeHome.
> |              |      Accuracy     |      AUROC      |
> |--------------|:-----------------:|:-----------------:|
> | DeepAll      |   21.75   |   54.07   |
> | UDG          |   19.91   |   55.95   |
> | DAML         |   29.35   |   55.55   |
> | FixMatch     |   40.68   |   57.25   |
> | OpenMatch    |   28.76   |  56.22   |
> | StyleMatch   |   42.57   |   56.72   |
> | CWAEE (ours) | **43.66** | **61.29** |
>
>
> W3: (Major) This paper evaluates their algorithm in only their setting. I think it can be evaluated in the same OOD setting in OpenMatch. (known/unknown in cifar10/cifar100 or Imagenet) If this paper could show the improvement in this setting, it would be more solid results.
>
> A: We focus on a more realistic but harder scenario where the learned model is not only required to classify *known classes* but also to recognize *unknown classes* on an *unseen* target domain. In order to carefully exploit unlabeled data from multiple source domains, we assign pseudo-labels to unlabeled samples whose scores are higher than the  *known classes* thresholds
>  $\delta^{1:|C^l|}\_{knw}$ or lower than the *unknown classes* thresholds $\delta^{1:|\mathcal{C}^l|}\_{unk}$. The unlabeled samples whose scores are between $\delta^{1:|\mathcal{C}^l|}\_{knw}$ and $\delta^{1:|\mathcal{C}^l|}\_{unk}$ are only utilized through consistency regularization loss $\mathcal{L}\_{con}^u$. In a degenerate situation where the model is trained and tested on the same domain, like the OOD setting in OpenMatch, OpenMatch applies entropy minimization on **all** unlabeled data which may be slightly more effective since it is an easier scenario to separate and utilize **all** unlabeled samples of *known* and *unknown classes*. However, in our setting, OpenMatch is less effective than ours as the experimental results shown in Table 1 in our paper.
>
> W4: (Minor) The one-vs-rest classifier is very similar to One-vs-All Outlier Detector in OpenMatch. (Novelty)
>
> A: Semi-Supervised Domain Generalization aims to use unlabeled data to improve the generalization of the model on *unseen* domains. Unfortunately, in a more realistic scenario, the unlabeled data usually contains *unknown classes* which may significantly degrade the generalization of the model, and the existing methods are not applicable to this scenario. The main contribution of our paper is that we propose a feasible solution with popularly used basic components, e.g. one-vs-all classifiers [4], class-wise adaptive thresholds [5] and consistency regularization [6], for this realistic scenario. The experiments conducted on realistic datasets verify the effectiveness and superiority of our method.
>
> [4] Rifkin et al., In Defense of One-vs-all Classification, JMLR'04.
>
> [5] Gui et al., Towards Understanding Deep Learning from Noisy Labels with Small-loss Criterion, IJCAI'21.
>
> [6] Sajjadi et al., Regularization with Stochastic Transformations and Perturbations for Deep Semi-supervised Learning, NIPS'16.

---

> > ### Comment · Reviewer_kd1R · 2023-08-21
> >
> > Thank you for the rebuttal. Many of my concerns are addressed. I am still a bit not convinced about training setting, but I appreciate the rebuttal. I raised my score.

---

> > > ### Author Response · Authors · 2023-08-21
> > >
> > > Thanks for your comments and kind suggestions. We will provide more discussions about the widely used training setting on semi-supervised domain generalization in the future revision.

---

> ### Author Response · Authors · 2023-08-16
>
> We sincerely thank you for your time and efforts in reviewing this paper and hope that our response has satisfactorily addressed your concerns. We are looking forward to discussing with you during the discussion period. We believe that your insights and suggestions can further improve this paper.

---

### Official Review · Reviewer_enwD · 2023-07-05

**Soundness:** 3 good
**Presentation:** 3 good
**Contribution:** 3 good
**Rating:** 7
**Confidence:** 3

**Summary:**

This paper focuses on the realistic scenario and proposes a semi-supervised domain generalization method. The method first explores unlabeled data by detecting known and unknown classes, and then exploits the data by adopting consistency regularization based on Fourier Transformation. The experiments show the effectiveness and superiority of the proposed method.

**Strengths:**

This paper is well-written and organized. It is easy to follow.
This paper focuses on the realistic scenario where the known classes are mixed with unknown classes in the data, and proposes a method which outperforms the state-of-the-art semi-supervised domain generalization methods.
This paper uses one-vs-rest classifiers and class-wise adaptive thresholds, which is helpful to detecting unknown classes.


**Weaknesses:**

The method needs to train one-vs-rest classifiers. When the number of classes is large, the computation cost is high.
The calibration needs the validation data.


**Questions:**

The method uses the class-wise adaptive thresholds to detect unknown classes. Why do these thresholds work intuitively?
There are hyper-parameters in Eq(9), is the performance sensitive to the hyper-parameters?

---

> ### Author Rebuttal · Authors · 2023-08-09
>
> W1: The method needs to train one-vs-rest classifiers. When the number of classes is large, the computation cost is high.
>
> A: When training $|\mathcal{C}^l|$ one-vs-rest classifiers for $|\mathcal{C}^l|$ *known classes*, we keep the architecture and parameters of the network unchanged and only replace the softmax function after $|\mathcal{C}^l|$-way linear classifier $h_\omega$ with $|\mathcal{C}^l|$ sigmoid function. These $|\mathcal{C}^l|$ one-vs-rest classifiers share the same backbone network $g_\theta$ and have their own classification heads $h_\omega^c$. Thus, the computation cost of $|\mathcal{C}^l|$ one-vs-rest classifiers in our method is actually equal to that of the softmax classifier.
>
> W2: The calibration needs the validation data.
>
> A: Score calibration, which uses the validation dataset, is a popularly used method for *unknown class* detection, *e.g.* [7, 8, 9, 10].
>
> [7] Guo et al., On Calibration of Modern Neural Networks, ICML'17.
>
> [8] Liang et al, Enhancing The Reliability of Out-of-distribution Image Detection in Neural Networks, ICLR'18.
>
> [9] Minderer et al, Revisiting the Calibration of Modern Neural Networks, NeurIPS'21.
>
> [10] Wang et al, Rethinking Calibration of Deep Neural Networks: Do Not Be Afraid of Overconfidence, NeurIPS'21.
>
>
> Q1: The method uses the class-wise adaptive thresholds to detect unknown classes. Why do these thresholds work intuitively?
>
> A: We train one-vs-rest classifiers for each *known class*, and the output scores of each one-vs-rest classifier indicate whether the sample belongs to the class or not. The samples of *unknown classes* do not belong to *known classes*, so the scores of *unknown classes* are usually lower than that of *known classes* on the corresponding one-vs-rest classifier. It is required to set proper thresholds for one-vs-rest classifiers to detect *known class* and *unknown classes*. However, due to the differences between the classes, it is difficult to manually choose the optimal threshold for each classifier. The score distributions of *known class* and *unknown classes* are usually different, as shown in Figure 4 in our paper. Thus, we use a two-component beta mixture model to fit the score distributions and set the thresholds with the means of two beta distribution components on each one-vs-rest classifier. The experiments conducted on realistic datasets also verify the effectiveness of our method.
>
> Q2: There are hyper-parameters in Eq(9), is the performance sensitive to the hyper-parameters?
>
> A: The performance of our method is not sensitive to the hyper-parameters. The results of the ablation study of hyper-parameters are in Appendix C of the submitted supplementary files.

---

> > ### Comment · Reviewer_enwD · 2023-08-16
> >
> > I have read the author's rebuttal and other reviews. The rebuttal has addressed my concerns. Therefore, I would like to keep my incline to accept the paper.

---

### Official Review · Reviewer_8khn · 2023-07-05

**Soundness:** 3 good
**Presentation:** 3 good
**Contribution:** 2 fair
**Rating:** 6
**Confidence:** 4

**Summary:**

Paper considers the setting of semi-supervised domain generalization (SSDG) when both unlabeled source and target domains can contain unknown classes, i.e. not seen as labeled instances in source domains. The goal here is to learn a classifier which will be able to (i) reliably distinguish seen classes from unknown classes and (ii) effectively utilize unlabeled source data to learn domain generalizable representations. To approach the aforementioned goals authors suggest using one-vs-rest classifiers and class adaptive thresholds to distinguish between known and unknown classes. And then they apply different optimization objectives for known/unknown classes to improve learning domain generalizable representations. The proposed methodology leads to improvements over considered baselines on the standard benchmarks.

**Strengths:**

-  The proposed problem formulation is important

**Weaknesses:**

-  Though the proposed setting is indeed realistic, the experimental setup is limited. It would be helpful to consider WILDS [1, 2] or other competitive benchmarks to understand the applicability of the proposed methodology.

- The proposed methodology looks to me like combination of many well-known techniques without thorough motivation and deep analysis.

[1] Pang Wei Koh et al' WILDS: A Benchmark of in-the-Wild Distribution Shifts. ICML 2021

[2] Shiori Sagawa et al' Extending the WILDS Benchmark for Unsupervised Adaptation. https://wilds.stanford.edu

**Questions:**

- How ImageNet pretraining affects the performance of the proposed method? What would be the performance if the method is trained from scratch?

**Limitations:**

Authors do not discuss limitations. One possible limitation that I see is that the proposed setting decides not to distinguish between different unknown classes. It would be interesting to push the limits of semi-supervised learning and study the ultimate setting when both domain/class shift are presented and the classifier is able to distinguish between different unknown classes. Some related papers include but not limited to [1, 2]

[1] Kaidi Cao, Maria Brbic, Jure Lescovec. Open-World Semi-Supervised Learning. ICLR 2022

[2] Sagar Vaze, Kai Han, Andrea Vedaldi, Andrew Zisserman. Generalized Category Discovery. CVPR 2022

---

> ### Author Rebuttal · Authors · 2023-08-09
>
> W1: Though the proposed setting is indeed realistic, the experimental setup is limited. It would be helpful to consider WILDS or other competitive benchmarks to understand the applicability of the proposed methodology.
>
> A: Thanks for your kind suggestions. The experiments of our paper are conducted on the datasets which are usually used within this research community [1, 2, 3], and the results verify the effectiveness of our method. We will conduct more experiments on the mentioned benchmarks.
>
> [1] Shu et al, Open Domain Generalization with Domain-Augmented Meta-Learning, CVPR'21.
>
> [2] Zhou et al, Domain Generalization with MixStyle, ICLR'21.
>
> [3] Zhang et al, Towards Unsupervised Domain Generalization, CVPR'22.
>
> W2: The proposed methodology looks to me like combination of many well-known techniques without thorough motivation and deep analysis.
>
> A: Semi-Supervised Domain Generalization aims to use unlabeled data to improve the generalization of the model on *unseen* domains. Unfortunately, in a more realistic scenario, the unlabeled data usually contains *unknown classes* which may significantly degrade the generalization of the model, and the existing methods are not applicable to this scenario. The main contribution of our paper is that we propose a feasible solution with popularly used basic components, e.g. one-vs-all classifiers [4], class-wise adaptive thresholds [5] and consistency regularization [6], for this realistic scenario. The experiments conducted on realistic datasets verify the effectiveness and superiority of our method.
>
> [4] Rifkin et al., In Defense of One-vs-all Classification, JMLR'04.
>
> [5] Gui et al., Towards Understanding Deep Learning from Noisy Labels with Small-loss Criterion, IJCAI'21.
>
> [6] Sajjadi et al., Regularization with Stochastic Transformations and Perturbations for Deep Semi-supervised Learning, NIPS'16.
>
> Q2: How ImageNet pretraining affects the performance of the proposed method? What would be the performance if the method is trained from scratch?
>
> A: Thanks for your kind suggestions. In order to verify the effectiveness of our method, we train the model from scratch with the compared methods and ours. The results on Officehome with 25:20:20 (*known classes*, *seen unknown classes* and *unseen unknown classes*) are summarized in Table 2. It can be found that our method outperforms other compared methods.
>
> Table 2: Leave-one-domain-out average results of *known classes* accuracy and *unknown classes* AUROC on OfficeHome.
> |              |      Accuracy     |      AUROC      |
> |--------------|:-----------------:|:-----------------:|
> | DeepAll      |   21.75   |   54.07   |
> | UDG          |   19.91   |   55.95   |
> | DAML         |   29.35   |   55.55   |
> | FixMatch     |   40.68   |   57.25   |
> | OpenMatch    |   28.76   |  56.22   |
> | StyleMatch   |   42.57   |   56.72   |
> | CWAEE (ours) | **43.66** | **61.29** |

---

> > ### Comment · Reviewer_8khn · 2023-08-14
> >
> > I thank the authors for their response. My concerns were partially addressed. I have updated my score to weak accept. Still, please do provide additional benchmark on more challenging datasets for the future revision. Providing the new setting requires deeper experimental study of the proposed methodology and its failure modes and will further strengthen the work.

---

> > > ### Author Response · Authors · 2023-08-15
> > >
> > > Thanks for your comments and kind suggestions. We will conduct more experimental studies on the proposed method in the new setting with additional datasets in the future revision.

---

### Official Review · Reviewer_mXoF · 2023-07-06

**Soundness:** 3 good
**Presentation:** 3 good
**Contribution:** 3 good
**Rating:** 7
**Confidence:** 4

**Summary:**

The paper considers the realistic semi-supervised domain generalization setting where known classes are mixed with some unknown classes in the unlabeled training and testing data, and proposes the Class-Wise Adaptive Exploration and Exploitation (CWAEE) method. The experiments conducted on the datasets show the advantages over the previous baselines.

**Strengths:**

1. Most previous semi-supervised domain generalization methods assumed that there is no unknown class in the unlabeled training data and testing data. The paper relaxes this assumption and considers a more challenging setting. The setting is new and important.

2. The paper proposes a two-step method. It first uses one-vs-rest classifiers and class-wise thresholds to detect the known and unknown classes, and then uses Fourier Transform and data augmentation to improve the generalization on target domain and unknown classes.

3. The method performs well on the datasets and the ablation study is sufficient to support the results.

**Weaknesses:**

1. Fourier Transform based on data augmentation may not work for non-image tasks.

2. Some techniques are similar to the well-known method FixMatch.

**Questions:**

My concerns for this paper are as follows:
1. Why does the method use the two-component beta mixture model to calculate the thresholds?

2. Why do the known classes have higher scores than the unknown classes? I think the authors should discuss more about this.

3. Will the number of unknown classes influence the performance? For example, the number of unknown classes is far larger than that of known classes in the unlabeled training data.


**Limitations:**

no negative societal impacts

---

> ### Author Rebuttal · Authors · 2023-08-09
>
> Q1: Why does the method use the two-component beta mixture model to calculate the thresholds?
>
> Q2: Why do the known classes have higher scores than the unknown classes? I think the authors should discuss more about this.
>
> Q3: Will the number of unknown classes influences the performance? For example, the number of unknown classes is far larger than that of known classes in the unlabeled training data.
>
> A: We train one-vs-rest classifiers for each *known class*, and the output scores of each one-vs-rest classifier indicate whether the sample belongs to the class or not. The samples of *unknown classes* do not belong to *known classes*, so the scores of *unknown classes* are usually lower than that of *known classes* on the corresponding one-vs-rest classifier. The score distributions of *known class* and *unknown classes* are usually different, as shown in Figure 4 in our paper. Thus, we use a two-component beta mixture model to fit the score distributions and set the thresholds with the means of two beta distribution components on each one-vs-rest classifier.
>
> Since we train the one-vs-rest classifier for each *known classes*, the number of *unknown classes* in unlabeled data will not influence the performance of our method on *known classes*, though the performance of our method on *unknown classes* may be influenced not significantly. We conduct more experiments on OfficeHome to verify the effectiveness of our method. We split the original label set into 5:25:20 and 5:50:10 (*known classes*, *seen unknown classes* and *unseen unknown classes*) to make the number of *unknown classes* far larger than that of known classes, and the results are summarized in Table 1. It can be found that the number of *unknown classes* does not influence the *known classes* accuracy of our method significantly, though the *unknown classes* AUROC drops slightly.
>
>
> Table 1: Leave-one-domain-out average results of *known classes* accuracy (left of the slash) and *unknown classes* AUROC (right of the slash) on  OfficeHome.
> |              |      5:25:20      |      5:50:10      |
> |--------------|:-----------------:|:-----------------:|
> | DeepAll      |   86.72 / 81.39   |   86.72 / 81.39   |
> | UDG          |   80.70 / 70.77   |   82.13 / 73.65   |
> | DAML         |   60.14 / 60.24   |   62.13 / 61.16   |
> | FixMatch     |   78.59 / 69.84   |   74.69 / 59.51   |
> | OpenMatch    |   88.20 / 82.31   |   86.88 / 82.37   |
> | StyleMatch   |   82.55 / 63.59   |   77.36 / 54.67   |
> | CWAEE (ours) | **89.40 / 86.69** | **88.44 / 83.80** |

---

### Decision · Program_Chairs · 2023-09-21

**Decision:**

Accept (spotlight)

**Comment:**

The paper studies an extension of semi-supervised domain generalization problem in which known classes can be mixed with unknown classes in unlabeled training and testing data.  In the proposed setting, the model needs to classify known classes and also recognize unknown classes. The authors propose a method for tackling this setting that is based on one-vs-rest classifiers and class-wise adaptive thresholds to detect known and unknown classes. Additionally, the authors propose consistency regularization on augmented training samples based on Fourier Transformation to improve generalization on an unseen domain.

Reviewers found that the problem setting tackled in this work is important and realistic, and the proposed method is effective. The paper  is well-written, well-motivated and easy to follow. The experimental results are convincing.

Some reviewers expressed concerns about the experimental design (limitedness of the benchmark datasets, model pretraining). During the rebuttal, the authors included additional results with training the model from scratch and promised to include more challenging benchmarks in the final version. Authors are strongly advised to include these experiments promised during the rebuttal.

Eventually all reviewers agree that the paper should be accepted. I therefore recommend acceptance.